# AcrIF9 tethers non-sequence specific dsDNA to the CRISPR RNA-guided surveillance complex

Marscha Hirschi[1,4], Wang-Ting Lu[2,4], Andrew Santiago-Frangos[3], Royce Wilkinson [3], Sarah M. Golden[3], Alan R. Davidson [2✉], Gabriel C. Lander [1✉] & Blake Wiedenheft [3✉]

Bacteria have evolved sophisticated adaptive immune systems, called CRISPR-Cas, that provide sequence-specific protection against phage infection. In turn, phages have evolved a broad spectrum of anti-CRISPRs that suppress these immune systems. Here we report structures of anti-CRISPR protein IF9 (AcrIF9) in complex with the type I-F CRISPR RNA-guided surveillance complex (Csy). In addition to sterically blocking the hybridization of complementary dsDNA to the CRISPR RNA, our results show that AcrIF9 binding also promotes non-sequence-specific engagement with dsDNA, potentially sequestering the complex from target DNA. These findings highlight the versatility of anti-CRISPR mechanisms utilized by phages to suppress CRISPR-mediated immune systems.

[1] Department of Integrative Structural and Computational Biology, The Scripps Research Institute, 10550 North Torrey Pines Road, La Jolla, CA 92121, USA. [2] Department of Molecular Genetics, and Department of Biochemistry, University of Toronto, 661 University Avenue, Room 1634, Toronto, ON M5G 1M1, Canada. [3] Department of Microbiology and Immunology, Montana State University, 1156 South 11th Avenue, Bozeman, MT 59717, USA. [4]These authors contributed equally: Marscha Hirschi, Wang-Ting Lu. ✉email: Alan.davidson@utoronto.ca; glander@scripps.edu; bwiedenheft@gmail.com

**B**acteria and archaea acquire immunity to viruses and other genetic parasites by preferentially integrating short fragments of foreign DNA into one end of a clustered regularly interspaced short palindromic repeat (CRISPR) locus[1–3]. CRISPR loci are transcribed and the long primary transcript is processed into a library of short CRISPR-derived RNAs (crRNA) that guide CRISPR-associated (Cas) proteins to complementary targets for nuclease-mediated degradation[4,5]. A diverse arsenal of CRISPR-Cas immune systems, comprising two classes, six types, and 32 subtypes, provide adaptive and heritable defense from genetic parasites[6,7]. In turn, phages and other parasites have evolved an equally varied repertoire of anti-CRISPR proteins[8]. Here we set out to determine the mechanism of immune system suppression by the previously identified, but mechanistically uncharacterized, anti-CRISPR protein IF9 (AcrIF9) that specifically inhibits the type I-F immune system in *Pseudomonas aeruginosa*[9]. We show that AcrIF9 inhibits the crRNA-guided surveillance complex (Csy) by sterically blocking the hybridization of target DNA to the crRNA guide. Interestingly, AcrIF9 bound to Csy, also promotes non-specific interactions with dsDNA, potentially sequestering the surveillance complex away from target DNA and thereby providing an additional layer of immune suppression.

## Results

### The cryo-EM structure of the Csy–AcrIF9 complex

In order to determine the mechanism of immune suppression by AcrIF9, we co-expressed and purified the Csy with AcrIF9 and determined the structure using cryo-electron microscopy (cryo-EM) at a nominal resolution of ~3.9 Å (Fig. 1, Supplementary Fig. 1, and Supplementary Table 1). Overall, the complex maintains the previously described seahorse-shape, comprising a hexameric Cas7f "backbone" that is capped on one end by the Cas6f "head", while the other end is capped by a heterodimeric Cas8f and Cas5f "tail". The Cas7f subunits are each shaped like a "right-hand" and two molecules of AcrIF9 (AcrIF9.1 and AcrIF9.2) bind the "thumbs" of the Cas7.4f and Cas7.6f subunits, respectively (Fig. 1a). The positioning of the AcrIF9 molecules is reminiscent of the anti-CRISPR AcrIF1 (Supplementary Fig. 2), which has previously been shown to sterically block access to the crRNA-guide for the complementary DNA target[10,11].

AcrIF9 is a 7.9 kDa protein made up of a five stranded, anti-parallel beta sheet cradling an alpha helix (Fig. 1b), a fold distinct from AcrIF1 (Supplementary Fig. 2). Both AcrIF9 molecules form extensive interactions with the thumbs of Cas7f (Fig. 1c, d), accounting for ~30% of the total solvent accessible surface area of AcrIF9. The two AcrIF9 binding sites are similar, and the specific residues interacting with each of the AcrIF9 molecules superimpose with an r.m.s.d. of 1.4 Å (see "Methods"). The contacts include a hydrogen bonding network between AcrIF9 residues Q11 to S15, and N94 and Q96 in Cas7f. In addition, Q11 of AcrIF9 also hydrogen bonds with a nucleobase from the crRNA (C13 to AcrIF9.1 and U25 to AcrIF9.2 respectively, Fig. 1c). A second cluster of hydrogen bonding interactions is found between AcrIF9 R17 and residues S89 and S92 in the Cas7f thumb (Fig. 1d). Furthermore, the aromatic sidechains of AcrIF9 residues F39, F40, and H41 form an "aromatic clamp" around residues L76 to T78 of Cas7f and additional hydrogen bonding occurs between AcrIF9 residues Q38 to H41 and Cas7f residues R75 to T78 (Fig. 1d). AcrIF9.1, but not AcrIF9.2, also interacts with Cas8f through hydrogen bonding of the Cas8f R224 carbonyl with AcrIF9.1

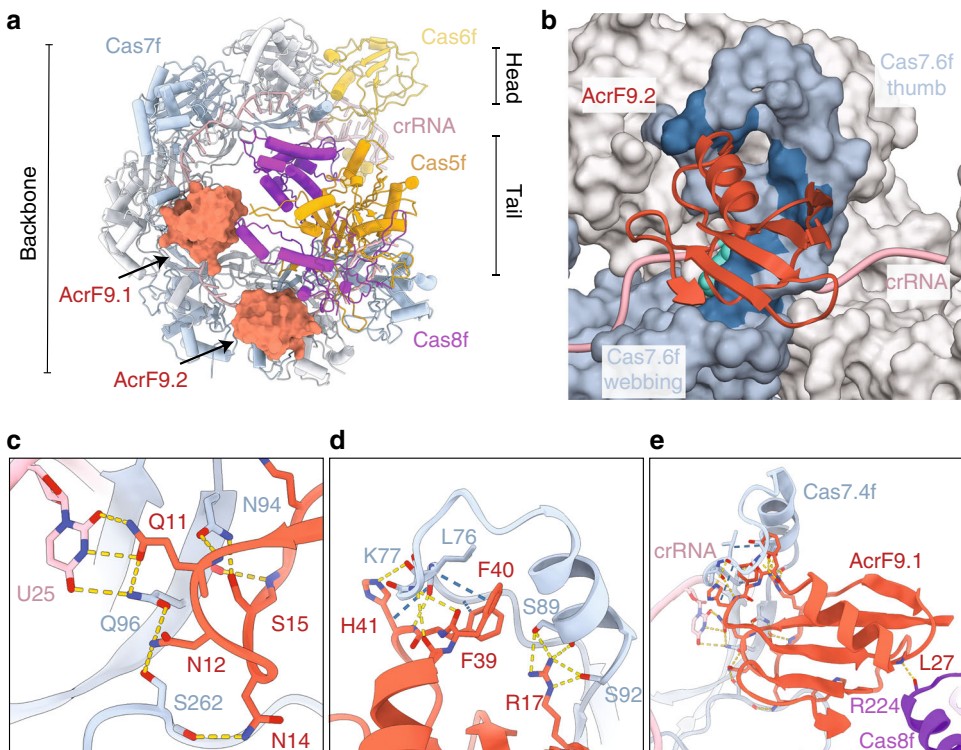

**Fig. 1 Cryo-electron microscopy structure of the Csy–AcrIF9 complex. a** Model of the Csy–AcrIF9 complex, Csy subunits shown as pipes and planks, Cas6f shown in yellow, Cas7f in light blue and gray, Cas8f in purple, Cas5f in orange and crRNA in pink, AcrIF9 shown in red surface representation. **b** The binding site of AcrIF9.2, residues interacting with AcrIF9 highlighted in blue (Cas7.6f) and green (crRNA). Cas7f subunits shown in surface representation, crRNA and AcrIF9 in cartoon representation. **c–e** Detailed view of the AcrIF9 binding site, model shown in cartoon representation, interacting residues shown as sticks. Interactions with uridine 25 (U25) are expected to accommodate any base at the equivalent position. Hydrogen bonds are indicated by yellow dashes, hydrophobic interactions are indicated by blue dashes.

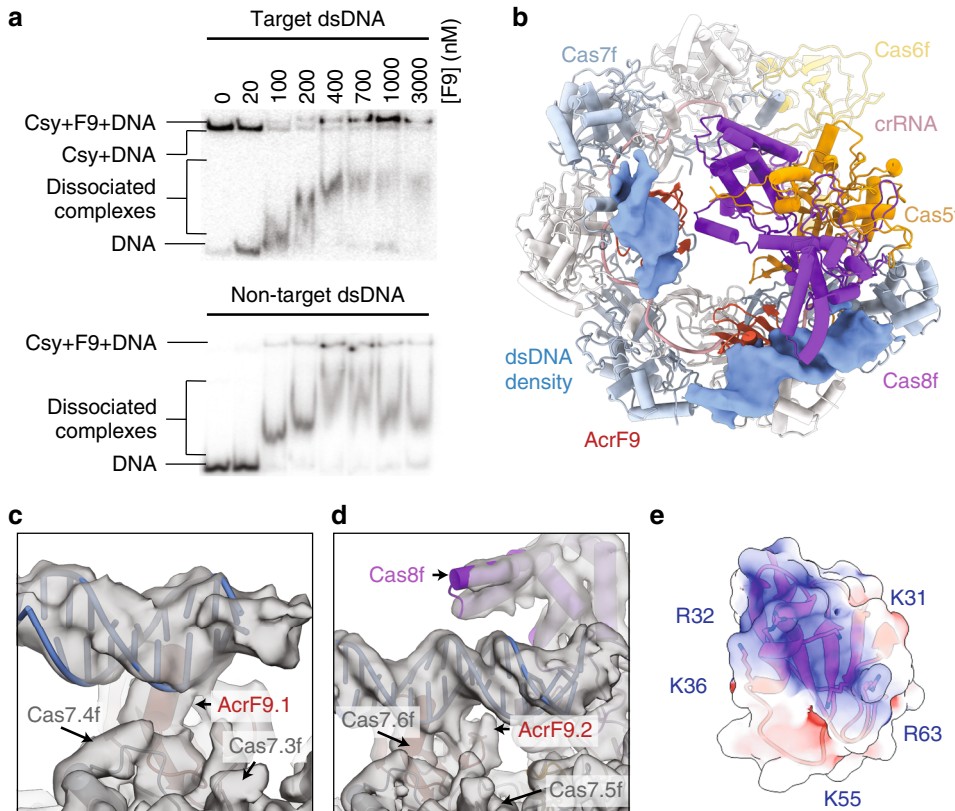

**Fig. 2 The Csy–AcrIF9 complex binds non-sequence-specific DNA. a** Csy–AcrIF9 binds target and non-target DNA as shown by EMSA. Csy (200 nM) is incubated for 15 min with increasing concentrations of AcrIF9 prior to the addition of [32]P-labeled target or non-target DNA. A proportion of the non-specifically bound DNA dissociates from Csy·AcrIF9 during gel electrophoresis. Data are provided as a Source Data file. **b** Model of the Csy–AcrIF9–dsDNA complex, Csy–AcrIF9 shown as pipes and planks with Gaussian-filtered EM density of the non-uniform refinement. **c**, **d** View of the Gaussian-filtered EM density of the non-uniform refinement at the AcrIF9.1 and AcrIF9.2 sites. **e** Electrostatic surface of the AcrIF9 DNA binding face, shown as a transparent surface, basic residues shown as sticks.

the L27 backbone nitrogen (Fig. 1e). While additional interactions between AcrIF9.1 and Cas8f are expected, the local resolution limits confident assignment of additional interactions at this interface.

**The Csy–AcrIF9 complex triggers non-specific DNA binding.** Based on the similarity of the binding sites for AcrIF1 and AcrIF9, we predicted that AcrIF9 would sterically block target binding. To test this hypothesis, we performed electrophoretic mobility shift assays (EMSA) to determine whether AcrIF9 is capable of inhibiting crRNA-guided interactions with target DNA (Fig. 2a). While target dsDNA (i.e., protospacer and PAM) binding is inhibited at sub-stoichiometric concentrations of AcrIF9 to Csy, dsDNA binding is unexpectedly restored at higher stoichiometric ratios of AcrIF9. EMSAs performed with non-target dsDNA (i.e., no protospacer) exhibited similar results (Fig. 2a). This result suggests that the formation of the ternary complex (i.e., Csy–AcrIF9–dsDNA) does not rely on base-pairing interactions and is fundamentally different than target DNA hybridization to the crRNA guide[12–14]. This binding behavior specifically requires the Csy–AcrIF9 complex, as AcrIF9 alone does not bind dsDNA (Fig. 2a and Supplementary Fig. 3).

**The cryo-EM structure of the Csy–AcrIF9–dsDNA complex.** To discern how interactions between AcrIF9 and Csy trigger binding to non-sequence-specific dsDNA, we next determined the structure of the Csy–AcrIF9 complex bound to dsDNA at a nominal resolution of ~4.2 Å (Fig. 2b, Supplementary Fig. 4, and Supplementary

Table 1). The reconstruction reveals two stretches of non-contiguous helical density that are not accounted for by Csy or AcrIF9 but are consistent with the size and shape of B-form DNA (Fig. 2b–d). The model reveals that both dsDNA segments are interacting with a positive patch on AcrIF9. Additionally, the dsDNA segment bound nearest Cas7.6f, interacts with residues from the Cas8f N-terminal hook (Fig. 2e). Notably, the positive patch on AcrIF9 is composed of five conserved residues (K31, R32, K36, K58, and R63) (Supplementary Fig. 5[15]). We attempted to make structure-guided mutations of these residues (AcrIF9[K31A,R32A,K36A], AcrIF9[K31Q,R32Q], and AcrIF9[K31E,R32E]). While the mutants express and purify similar to wild-type AcrIF9, the size exclusion profiles are distinct (Supplementary Fig. 6). In addition, the mutants are defective for blocking crRNA-guided interactions with DNA targets. Collectively, these observations suggest the mutations result in a folding defect that reduces the affinity for Csy.

## Discussion

Collectively, the structures and biochemistry presented here suggest a two-pronged mechanism for AcrIF9-mediated immune suppression. Similar to AcrIF1, AcrIF9 inhibits the surveillance system by sterically blocking crRNA-guided binding to complementary DNA. While this manuscript was under review, Zhang et al. published a Csy–AcrIF9 structure and similarly concluded that AcrIF9 inhibits target-DNA hybridization to the crRNA guide[16]. The two Csy–AcrIF9 models are nearly identical and align with an all-atom r.m.s.d. of 3.7 Å. Here, we provide

additional insights into AcrIF9 function and show how the Csy–AcrIF9 complex promotes the binding of dsDNA in a non-sequence-specific manner. A similar phenomenon has recently been reported for AcrIIA11, an anti-CRISPR targeting Cas9[17], although structural information is currently unavailable. We suggest that Acr-triggered interactions with non-sequence-specific dsDNA could represent an additional layer of immune suppression by sequestering the surveillance system through non-productive associations with dsDNA. However, future experiments are required to reveal the importance and prevalence of Acr-induced non-specific DNA sequestration.

## Methods

**Purification of the Csy complex.** Csy genes and a synthetic CRISPR were co-expressed on separate vectors (Addgene IDs: 89232 and 89244) in *Escherichia coli* BL21 (DE3) cells (NEB)[18,19]. The expression was induced with 0.5 mM isopropyl-D-1-thiogalactopyranoside (IPTG) at an optical density (OD$_{600\,nm}$) of 0.5. Cells were incubated overnight at 16 °C, then pelleted by centrifugation (5000 × g for 15 min at 4 °C) and re-suspended in lysis buffer containing 50 mM 4-(2-hydroxyethyl)-1-piperazineethanesulfonic acid (HEPES) at pH 7.5, 300 mM potassium chloride, 5% glycerol, 1 mM Tris(2-carboxyethyl) phosphine hydrochloride (TCEP), and 1× protease inhibitor cocktail (Thermo Scientific). Pellets were sonicated on ice for 3 × 2.5 min (1 s on followed by 3 s off). The lysate was clarified by centrifugation at 22,000×g for 30 min at 4 °C. The Csy complex self-assembles in vivo and the His-tagged complex (N-terminal 6-histidine affinity tags on Cas7f) was affinity purified using Ni-NTA resin (QIAGEN). The resin was washed with five column volumes of lysis buffer supplemented with 20 mM imidazole before elution with lysis buffer supplemented with 300 mM imidazole. Protein was then concentrated (Corning Spin-X concentrators) at 4 °C prior to size-exclusion chromatography (Superdex 200, GE Healthcare) in 20 mM HEPES, pH 7.5, 100 mM KCl, 5% glycerol, and 1 mM TCEP.

**AcrIF9 purification.** AcrIF9 from *Proteus penneri* (EEG86164.1) was cloned into pMAL-c4x, downstream of an N-terminal maltose binding protein (MBP) and an HRV-3C protease site (Addgene ID: 141442). Primer sequences used for cloning are as follows; forward TTCCAAGGTCCTATGAAAAGCACATACATCATC, and reverse AAGAACTTCAAGGAATTCTGAAATCCTTCCC. The expression plasmid was transformed into *E. coli* BL21 (DE3) cells and protein expression was induced with IPTG at an optical density (OD$_{600\,nm}$) of 0.5. Cells were pelleted and lysed as described above. AcrIF9 was purified in lysis buffer (50 mM Tris, pH 7.5, 300 mM NaCl, 5% glycerol), which was supplemented with 10 mM maltose for elution off MBP resin (GE Healthcare). The protein was concentrated (Corning Spin-X concentrators) at 4 °C prior to size-exclusion chromatography (Superdex 75, GE Healthcare) in 20 mM Tris, pH 7.5, 100 mM NaCl, and 5% glycerol. The MBP tag was removed with HRV-3C protease and the tag was separated from AcrIF9 using size-exclusion chromatography (Superdex 75, GE Healthcare) in 20 mM Tris, pH 7.5, 100 mM NaCl, and 5% glycerol.

**Purification of Csy–AcrIF9 complex.** Csy genes (Addgene ID: 89232), a synthetic CRISPR (Addgene ID: 89244), and AcrIF9 (GenBank: EEG86164.1 cloned into pCDF-1b) were co-expressed on separate vectors, as described above. The Csy–AcrIF9 complex was purified using methods similar to those described for the Csy complex, except size exclusion was performed using a Superdex 200 26/600 column (GE Healthcare), equilibrated in 20 mM HEPES, pH 7.5, 100 mM NaCl, 5% glycerol, and 1 mM TCEP.

**Purification of Csy–AcrIF9–dsDNA complex.** Purified Csy was first incubated with a 4-fold excess of AcrIF9 in buffer (20 mM HEPES, pH 7.5, 100 mM NaCl, 5% glycerol, and 1 mM TCEP) at 37 °C for 15 min. Purified non-target dsDNA (described below) was then added at a 2.5-fold excess over Csy and incubated for an additional 15 min at 37 °C. Csy–AcrIF9–dsDNA complex was then purified using a Superdex 200 10/300 size exclusion column (GE Healthcare) and the sample was concentrated using a 100 kDa MWCO concentrator (Pierce).

**Nucleic acid preparation.** All ssDNAs were purchased from Eurofins Genomics and purified by reverse-phase high-performance liquid chromatography. Specific (5′-GCTGTACGTCACTATCGAAGCAATACAGGTAGACGCGGACATCAAG CCCGCCGTGAAGGTGCAGCTTCTCTACAGAGTGC-3′) and non-specific (5′-GCAGCTCGAGTTAAGACGGTATTGTTCAGATCCTGGCTTGCCAACAGTG ATTTGCTCATTTTGTAGATTGAGTCGCT-3′) DNA targets (1 pmol) were labeled at the 5′-end with 2 pmol of [γ-$^{32}$P]ATP (PerkinElmer), using T4 poly-nucleotide kinase (NEB) in 1× polynucleotide kinase buffer at 37 °C for 45 min[20]. Polynucleotide kinase was heat-denatured by incubation at 65 °C for 20 min. Unincorporated [γ-$^{32}$P]ATP was removed from the reaction using gel filtration spin columns (G-25, GE Healthcare). $^{32}$P-labeled ssDNAs were mixed with a 10-fold excess of unlabeled complementary ssDNA, in 20 mM Tris–HCl, pH 7.5, 100 mM KCl, 5 mM MgCl$_2$. DNA mixtures were denatured at 95 °C for 5 min, and

then annealed by slow cooling to room temperature over 1 h. $^{32}$P-labeled dsDNAs were purified by native gel electrophoresis, ethanol precipitated, and resuspended in 10 mM Tris–HCl, pH 7.5, 50 mM NaCl, 1 mM ethylenediaminetetraacetic acid.

**Electrophoretic mobility shift assay.** Binding assays were performed by incubating increasing concentrations of AcrIF9 with 200 nM Csy complex in reaction buffer (20 mM HEPES, pH 7.5, 100 mM potassium acetate, 5% glycerol, 1 mM TCEP) for 15 min at 37 °C. Reactions were moved to ice and <0.5 nM of 5′ $^{32}$P-labeled DNA oligonucleotides were added. Reactions were then incubated for 15 min at 37 °C. Reaction products were run on native, 6% polyacrylamide gels, which were dried and imaged with a phosphor storage screen (Kodak), then scanned with a Typhoon phosphorimager (GE Healthcare).

**Electron microscope sample preparation.** Csy–Acr complexes were previously found to exhibit preferred orientation when suspended in open holes during preparation for cryo-EM imaging[10]. Therefore, purified Csy–AcrIF9 was mixed with 0.05% v v$^{-1}$ Lauryl Maltose Neopentyl Glycol prior to freezing. UltrAuFoil R1.2/1.3 300-mesh grids (Electron Microscopy Sciences) were plasma cleaned immediately prior to sample application in a Solarus plasma cleaner (Gatan, Inc.) with a 75% nitrogen, 25% oxygen atmosphere at 15 W for 7 s. Cryo-EM grids were prepared by application of 4 μL Csy–AcrIF9 at a concentration of 2.5 mg mL$^{-1}$. Grids were manually blotted with Whatman No. 1 filter paper for 5 s followed by plunge freezing in liquid ethane at 4 °C in 95% humidity.

**Cryo-EM data acquisition.** Cryo-EM data was collected using the Leginon[21] automated data collection software on a Talos Arctica (Thermo Fisher) TEM operating at 200 keV equipped with a K2 Summit direct electron detector (Gatan, Inc.) in counting mode. Movies were collected at a nominal magnification of 36,000× with a physical pixel size of 1.15 Å pixel$^{-1}$. A total of 1860 movies were collected with a 200 ms frame rate and a total exposure time of 13.1 s. An exposure rate of ~5 electrons pixel$^{-1}$ s$^{-1}$ was used, resulting in a cumulative exposure of 66 electrons Å$^{-2}$. Data were acquired using a nominal defocus range of 0.8–1.2 Å. To improve the Euler distribution, grids were tilted to 30° during imaging[22]. Pre-processing was performed in real-time to monitor data quality using the Appion image processing pipeline[23]. Frame alignment, CTF estimation, and particle picking were performed with MotionCorr2[24], Gctf[25], and DoGpicker[26], respectively.

Data collection on the Csy–AcrIF9–dsDNA sample was performed as described above with the following exceptions: a total of 3413 movies were collected tilted to 30° and 3059 images were collected untilted with a total exposure time of 12 s and 100 ms frame rate.

**Csy–AcrIF9 cryo-EM data processing.** Movies of the Csy–AcrIF9 complex were aligned and dose-weighted using Motioncorr2 in Relion 3.0 on 5 × 5 frames with an applied B-factor of 150[27]. Unweighted aligned images were used for CTF estimation in Gctf[25]. Difference of Gaussian[26] picker was used to pick particles from the first 1017 dose-weighted aligned images and particles were extracted binned 4 × 4 (4.6 Å pixel$^{-1}$, 48-pixel box size). Reference-free 2D classification was performed on the particle stack to generate templates. One top view and one tilt view were selected as templates and used for template picking of all movies, resulting in a total of 1.57 million picks that were extracted binned 4 × 4 (4.6 Å pixel$^{-1}$, 48-pixel box size). The particle stack was subjected to reference-free 2D classification and non-particle picks were eliminated from the stack. This particle stack of 1.53 million particles was input to 3D auto-refinement. An initial model was obtained from a stack generated by Appion during data collection of the first 500 images. The stack was uploaded to CryoSparc[28] and subjected to 2D classification. 2D averages showing secondary structure elements were selected and used to generate an ab initio 3D model. This model was imported into Relion and used as a reference for 3D auto-refinement of the 1.53 M particle stack. The coordinates of the particles were re-centered based on the alignment parameters of the reconstruction, and the particles were re-extracted at a binning of 2 × 2 (2.3 Å pixel$^{-1}$, 96-pixel box size) and subjected to 3D classification (3 classes, tau-value 10, performing angular and translational searches). The 314,854 particles contributing to the one high-resolution 3D class were re-centered and re-extracted at full resolution (1.15 Å pixel$^{-1}$, 192-pixel box size). The unbinned stack refined to a nominal resolution of 6.6 Å using a mask generated by converting the Csy–AcrIF1/AcrIF2 structure (PDB ID: 5uz9) into a low-resolution density map using the "molmap" function in Chimera[29], and then subsequently applying a 4-pixel extension and 8-pixel soft cosine edge using Relion. The resulting reconstruction was low-pass filtered to 15 Å for use in generating a new soft mask with a 4-pixel extension and 8-pixel soft cosine edge. The particles were grouped by image shift and iteratively CTF refined with beam tilt and per-particle astigmatism correction followed by auto-refinement. Four rounds of CTF refinement and auto-refinement led to a reconstruction with nominal resolution of 4.3 Å. Subsequent particle polishing and a final round of CTF refinement resulted in an improved map with a resolution of ~3.9 Å. The local resolution was estimated using the blocres program from the Bsoft package[30].

**Csy–AcrIF9–dsDNA cryo-EM data processing.** Movies of the Csy–AcrIF9–dsDNA complex were aligned and dose-weighted in the Appion pipeline using Motioncorr2 on 5 × 5 frames with an applied B-factor of 100. The aligned and dose-weighted

micrographs were imported into Cryosparc v2.14.2 and patch CTF estimation was performed. One top view, one side view, and one tilt view from the Csy–AcrIF9 dataset were selected as templates and used for template picking of all micrographs, resulting in a total of 1.91 million picks that were extracted unbinned (1.15 Å pixel$^{-1}$, 192-pixel box size). Particle pick inspection was performed to remove false picks or damaged, denatured, or aggregated particles, resulting in a stack of 1.05 million particles. This particle stack was input to 3D heterogeneous refinement with 4 classes, a reconstruction of the Csy complex (without Acr bound) was used as an input model. One resulting class with high-resolution details, containing 505,563 particles, was selected for further processing. Two subsequent rounds of heterogeneous refinement with 3 classes were performed using the previous result as input model and selection of the highest resolution class. After the third round, heterogeneous refinement did not further benefit the quality and resolution of the reconstruction. This particle stack, containing 152,066 particles, was refined using non-uniform refinement to a reported resolution of 4.2 Å. The resulting particle stack was imported into Relion 3.1. The Cryosparc reconstruction was low-pass filtered to 15 Å for use in generating a new soft mask with a 4-pixel extension and 8-pixel soft cosine edge. The stack was auto-refined to a reconstruction with a nominal resolution of 4.5 Å. The particles were grouped by image shift and iteratively CTF refined with beam tilt and per-particle astigmatism correction followed by auto-refinement. Ctf refinement followed by auto-refinement led to a reconstruction with a nominal resolution of 4.2 Å. Subsequently, the local resolution was estimated as described above using Bsoft[30].

**Atomic model building and refinement**. The atomic model was built and real-space refined in Coot constrained to ideal geometry and secondary structure, where appropriate. The model of Csy–AcrIF1/AcrIF2 structure (PDB ID: 5uz9), stripped of AcrIF1/AcrIF2, was rigid body fit into the EM-density map as a whole and subsequently by chain. The model was refined into the density with application of Geman–McClure distance restraints generated in ProSMART[31]. The model was manually inspected and adjusted where required, specifically loop regions of the Cas7f thumb, web and residues 291–296 were rebuilt. Sidechains were removed where supporting cryo-EM density was lacking, most notably in the regions of the head and the helical bundle of Cas8f.

The model for AcrIF9 was built de novo by placing idealized helices and ß-strands in the EM density. Loops were built into the density connecting the secondary structure elements. Bulky sidechains were built and used to determine the register of the model. Sidechains were built using optimal rotamer conformations as indicated by the cryo-EM density.

In order to build the atomic model for Csy–AcrIF9–dsDNA, the Csy–AcrIF9 model, the Cas8f N-terminal hook from the Csy–AcrIF1/AcrIF2 structure (PDB ID: 5uz9) and idealized B-form DNA were rigid body fit into the EM-density map. The model was refined into the density with application of Geman–McClure distance restraints generated in ProSMART[31]. As the density does not allow for the identification of the DNA bases the model was trimmed to backbone atoms.

The models were iteratively real-space refined by global minimization and rigid body refinement using the Phenix[32] command line and Coot[33], each adhering to ideal geometry and secondary structure using Ramachandran and secondary structural constraints in Phenix. The model was further optimized for compliance to geometric constraints using Molprobity[34]. Figures were rendered in ChimeraX[35].

In order to compare the AcrIF9 binding sites, the AcrIF9 molecules were superimposed using Pymol followed by calculation of the all-atom r.m.s.d. between the Csy residues within 4 Å from AcrIF9.1 or AcrIF9.2. The residues within 4 Å from AcrIF9.1 are Cas7.4f 72–78, 85–86, 89, 92–96, and 258–262 and crRNA residues 24 and 25, the residues within 4 Å from AcrIF9.2 are Cas7.6f 72–78, 85–86, 89, 92–96, and 258–262 and crRNA residues 12 and 13.

**Reporting summary**. Further information on research design is available in the Nature Research Reporting Summary linked to this article.

## Data availability

The cryo-EM maps of Csy–AcrIF9 and Csy–AcrIF9–dsDNA have been deposited in the Electron Microscopy Data Bank (EMDB) under the accession codes EMD-21516 and EMD-21517, respectively. Atomic coordinates of the models have been deposited in the Protein Data Bank (PDB) under accession codes 6W1X and 6WHI. Data for Fig. 2a and Supplementary Figs. 1a, 3, 4a, and 6b are provided as a Source Data file. Other data are available from the corresponding authors upon request.

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

## Acknowledgements
The authors are grateful to Bill Anderson, The Scripps Research Institute (TSRI) Electron Microscopy facility manager, and Jean-Christophe Ducom at TSRI High Performance Computing for support during EM data collection and processing. The authors thank MaryClare Rollins and the rest of the Wiedenheft and Lander labs for critical feedback and discussion and Jack Nunberg and Ewelina Malecka-Grajek for generous assistance with imaging. G.C.L. is supported as a Searle Scholar, a Pew Scholar, by a young investigator award from Amgen, and the National Institutes of Health (DP2EB020402). M.H. is supported by the Dennis and Marsha Dammerman Fellowship of the Damon Runyon Cancer Research Foundation. A.S.-F. is a postdoctoral fellow of the Life Science Research Foundation, which is supported by the Simons Foundation. Research in the Wiedenheft lab is supported by the National Institutes of Health (R35GM134867), a young investigator award from Amgen, the M. J. Murdock Charitable Trust, and the Montana State University Agricultural Experimental Station (USDA NIFA). Computational analyses of EM data were performed using shared instrumentation at TSRI funded by NIH S10OD021634.

## Author contributions
M.H. conducted the electron microscopy experiments, single-particle 3D reconstruction, and model building under the guidance of G.C.L. W.-T.L. observed the non-specific DNA binding capacity triggered by AcrIF9 under the guidance of A.R.D. A.S.-F., R.W., and S.M.G. purified the protein samples and performed in vitro biochemical experiments under the guidance of B.W. M.H., A.S.-F., R.W., G.C.L., and B.W. wrote the manuscript.

## Competing interests
B.W. is the founder of SurGene, LLC, and is an inventor on patent applications related to CRISPR-Cas systems and applications thereof. A.R.D. is a scientific advisory board member for Acrigen Biosciences and is an inventor on patents relating to anti-CRISPR proteins. The other authors declare no competing interests.
