## [Peer Review File · Nature Communications]

Reviewers' comments:

Reviewer #1 (Remarks to the Author):

This short paper reports on the mechanism of the anti-CRISPR AcrIF9 (F9), which inhibits the type I-F (Csy) effector. This Acr was one of the first identified but has not been characterised structurally or mechanistically. This is achieved here using a combination of cryo-EM and some basic biochemical analyses. The Cryo structure reveals that AcrIF9 binds to the type I-F complex as two monomers by interacting with the Cas7 subunit in a manner that will block target RNA search and binding – a mechanism remarkably similar to AcrIF1.

As judged by EMSA, AcrIF9 does not bind dsDNA on its own up to a concentration of 3 μ M, but does inhibit Csy binding to target DNA when present at 20 nM concentration (ten-fold less than the Csy concentration) – see point 3 below. However, at higher stoichiometries, F9 no longer prevents binding to DNA, which is also difficult to explain.

A cryo structure was also determined for the complex incubated with dsDNA, revealing binding of the DNA to the surface of F9, allowing mapping of the interface as a basic area of F9 with conserved basic residues (see point 5). The authors speculate that F9 functions partly by sequestering Csy on random DNA as well as blocking target DNA binding.

Overall there is some decent cryo-EM data here paired with some preliminary, contradictory and variable quality EMSA data. This leaves any conclusion beyond "F9 probably functions like F1" to be on rather shaky ground.

Specific points:

1. AcrIF9 is described as a 5-stranded anti-parallel beta sheet cradling an alpha helix, but no structural comparison is offered. Are there any similar folds in PDB? Can the structure be compared more fully to AcrIF1, given their striking convergence in binding sites? Can the two binding sites be compared, beyond saying they are "similar".
2. The "triplicate" EMSAs shown in figure S3 are strikingly different from one another to a surprising degree. For example, the third gel in S3b is quite fundamentally different from the first gel. In the second and third replicates it looks like there is a significant proportion of shifted material half way up the gel over a wide protein range, suggesting an intermediate species, but the gel in figure 2 doesn't conform to this. These experiments should be repeated to give more confidence in the data and their interpretation. Ideally, an alternative technique such as fluorescence polarization should be considered, as binding or displacement of Csy will give rise to a large anisotropy change using a method that measures true equilibria.
3. Figure 2a suggests that one tenth molar ratio of AcrIF9 almost completely blocks binding of Csy to target DNA. Given that at least 90% of the Csy complex should be free and available to bind DNA (assuming non-cooperative binding of AcrIF9), what is the authors' view of the mechanism here? Notably, in the replicates, this effect is not observed at 20 nM F9 to nearly the same extent, and in the middle gel of S3a there appears to be 50:50 binding to target DNA at a molar ratio of 1:2, which could make more sense.
4. It would help to understand how Csy binds to target and non-target DNA in the absence of F9. Although this has no doubt been published previously it would be helpful to show this data for the DNA species used here. It is important to understand how F9 affects DNA binding for both target and non-target DNA.
5. A binding interface for F9 on dsDNA is observed that does not involve the Csy complex. The text states that "we attempted to make structure-guided mutations of these residues, but the F9 mutants failed to bind the Csy complex". Leaving aside the observation that there is no data in the paper to support this (were all 5 made? Did they purify as for the wild-type protein?), this is another puzzling observation as the binding interface between F9 and Csy has not been modified.

Reviewer #2 (Remarks to the Author):

Hirschi et al. present the cryo-EM structures of Csy-AcrIF9 complex and Csy-AcrIF9-dsDNA complex at $\sim 3.9 \text{ \AA}$ and 6.9 \AA respectively. In combination with biochemical data, the authors show AcrIF9 not only sterically blocks the binding of target DNA to crRNA, but also promotes nonspecific recruitment of dsDNA, potentially sequestering the complex from target DNA. Even without the high-resolution structure of Csy-AcrIF9-dsDNA complex, the biochemical data and the current density map support their conclusions. This paper significantly broadens the current view of the mechanisms for anti-CRISPR proteins mediated immune suppression, and will be of interest in the community given that CRISPR-Cas immunity is of broad interest.

During the revision session, the structure of Csy-AcrF9 complex has been described in a PNAS paper (PNAS first published March 13, 2020 <https://doi.org/10.1073/pnas.1922638117>). With 2.57 \AA high-resolution Csy-AcrF9 complex structure, the authors of the PNAS paper proposed binding of AcrF9 to the Csy complex prevents target DNA binding to crRNA, similar to how AcrF1 works, also in line with part of Hirschi et al.'s conclusions. The author may briefly discuss the results in the PNAS paper. Even the high-resolution structure of Csy-AcrF9 complex recently has been reported, as far as I see, the non-sequence specific dsDNA bound to Csy-AcrF9 complex reported in Hirschi et al. paper significantly advances the understanding of how anti-CRISPR proteins work, I still support this paper to be published.

Minor comments:

- 1) Fig. 1b, the color of Cas7.6f thumb label is not in blue, making the label not clear that what it represents.
- 2) The authors should provide detailed information in the method session about how Csy-AcrIF9 and Csy-AcrIF9-dsDNA complexes are assembled.
- 3) By comparing Supplementary Fig.1a and Supplementary Fig.4a, it's confusing about the peaks they choose. The authors might use different columns to purify the complex, are the two major peaks in each file related to each other in these two gel filtration files?
- 4) In supplementary Fig, 4a, the label "10, 11, 13, 15" is not clear about what it presents.

Reviewer #3 (Remarks to the Author):

The authors present structural and biochemical analyses of the anti-CRISPR protein AcrIF9 that targets the type I-F CRISPR-Cas system. They have determined the 3D structure of AcrIF9 bound to the RNA-guided surveillance complex (Csy) by cryo-EM at an overall resolution of 3.9 \AA . Two copies of AcrIF9 bind to the Csy complex at positions reminiscent of the AcrIF1 binding sites. AcrIF9, like AcrIF1, blocks target DNA binding in Electrophoretic Mobility Shift Assays with sub-stoichiometric ratios of AcrIF9 to Csy complex. However, at higher stoichiometric ratios, AcrIF9 induces target and non-target DNA binding to Csy. The 3D reconstruction of the Csy-AcrIF9-dsDNA ternary complex, determined by cryo-EM at an overall resolution of 6.9 \AA , shows that AcrIF9 tethers dsDNA to the Csy complex.

As acknowledged by the authors, a similar phenomenon of anti-CRISPR-induced non-specific DNA binding to CRISPR-Cas surveillance complex has previously been reported for AcrIIA11 that targets the type II CRISPR-Cas9 system (<https://doi.org/10.7554/eLife.46540>). Although the manuscript under review presents the first example of a type I anti-CRISPR protein promoting non-specific DNA binding to the Csy complex in vitro, major revisions are required to strengthen the authors

findings and provide an advance in understanding anti-CRISPR proteins. In particular, the role for dsDNA binding in AcrIF9's mode of inhibition remains elusive. What would be the benefit of a concentration-dependent two-pronged CRISPR-Cas inhibition mechanism while the inhibition of target DNA binding and cleavage is effective at low concentration of anti-CRISPR protein (sub-stoichiometric to stoichiometric ratios of AcrIF9 to the Csy complex)? Noteworthy, the paper by Zhang K et al. entitled "Inhibition mechanisms of AcrF9, AcrF8, and AcrF6 against type I-F CRISPR-Cas complex revealed by cryo-EM", which has been published in the course of reviewing this manuscript (www.pnas.org/cgi/doi/10.1073/pnas.1922638117), presents a highly detailed cryo-EM structure of AcrIF9 bound to the Csy complex at an overall resolution of 2.6 Å and shows that AcrIF9 functions via the competitive binding to DNA binding sites on Cas7f subunits. Overall, further data interpretation and additional experiments could greatly improve the manuscript. Besides, all methodological details should be provided for a better understanding of the results and for the sake of reproducibility.

Major comments

1) Page 3: "To achieve this goal, we co-expressed and purified the crRNA-guided surveillance complex (Csy) with AcrIF9 and determined the structure [...]"

- Here one understands that AcrIF9 together with the crRNA-guided surveillance complex were co-expressed. However, there is no paragraph related to this procedure in the Methods. Alternatively, if AcrIF9 and the Csy complex have been produced independently, as described in the Methods, and then mixed to make the complex, this procedure should be described in the Methods. This point should be clarified.

2) Fig 1, panels d-e; page 4: "The two AcrIF9 binding sites are nearly identical. Residues interacting with each of the AcrIF9 molecules superimpose with an r.m.s.d. of 1.4 Å (see methods). The contacts include: a hydrogen bonding network involving AcrIF9 residues 11 to 15, a β-sheet on the Cas7f thumb, and a nucleobase from the crRNA (Fig. 1c); interactions between AcrIF9 arginine 17 and residues in the Cas7f thumb (Fig. 1d); and the formation of an "aromatic clamp" around the Cas7f thumb loop by AcrIF9 residues 39 to 41 (Fig. 1d). AcrIF9.1, but not AcrIF9.2, also interacts with Cas8f (Fig 1e)."

-This description of the intermolecular contacts between AcrIF9 and the Csy complex should be revised for a better reading and understanding.

- The authors mention an "aromatic clamp" formed by AcrIF9 residues, which suggests hydrophobic or stacking interactions between AcrIF9 and Cas7f residues. However, panel d shows that these aromatic residues hydrogen bond with Cas7f.

- The authors mention "interactions between AcrIF9 arginine 17 and residues in the Cas7f thumb". The term "interactions" is vague. According to panel d, these interactions are hydrogen bonds, as are all intermolecular contacts described in this section.

- The authors indicate that the two AcrIF9 binding sites are nearly identical. Is the only difference the interaction with Cas8f? This interaction between Cas8f and AcrIF9.1, shown as one hydrogen bond in panel e, should also be described in the text.

- The numbering of all interacting residues and nucleobases should be reported in the text and the figure panels.

3) Page 5: "Based on the similarity of the binding sites for AcrIF1 and AcrIF9, we predicted that AcrIF9 would sterically block target binding."

-The authors have previously shown that AcrIF1 interacts with Cas7f lysine residues (K85, K254, K257) that are involved in target DNA binding (<http://dx.doi.org/10.1016/j.cell.2017.03.012>). Based on Fig 1 panel d, the lysine 85, which is located in the thumb of Cas7f, may also interact with AcrIF9. Testing the binding of AcrIF9 to wild type Csy and to the K85A mutant used in the paper mentioned above, could experimentally prove that AcrIF9 is a steric inhibitor of target DNA binding.

4) Page 6: Fig 2, panel A

- In the EMSA with target dsDNA, the strong unbound DNA band in the 20 nM lane is puzzling. Indeed, it indicates that most of the Csy molecules are inhibited, which cannot be the case with a steric inhibitor at sub-stoichiometric concentration. The second EMSA presented in the Supplementary Fig 3 panel a would be more appropriate for the main figure.
- In the EMSA with target and non-target dsDNAs, how do you interpret the DNA migration profiles of the 200 nM lanes? At 200 nM of AcrIF9, and with two AcrIF9 binding sites per Csy molecule, not all Csy molecules are fully occupied by the inhibitor.
- Overall, how do you interpret these EMSA in terms of the AcrIF9 molecular mechanism? If a Csy-mediated DNA cleavage assay in the presence of a stoichiometric ratio of AcrIF9 shows an inhibition of the Csy activity (which is not the case for AcrIIA11), what would be the benefit of binding dsDNA at high concentrations of AcrIF9?

5) Page 5: "we next determined the structure of the Csy-AcrIF9 complex bound to dsDNA at a nominal resolution of $\sim 6.9 \text{ \AA}$ (Fig. 2b and Supplementary Fig. 4)."

- The procedure used to make the Csy-AcrIF9-dsDNA ternary complex is not described in the Methods.

6) Page 5: "The reconstruction reveals two stretches of non-contiguous helical density that are not accounted for by Csy or AcrIF9, but are consistent with the size and shape of B-form DNA (Fig. 2c)."

- The EMSA with target and non-target dsDNAs show that non-specific DNA binding occurs at 400 nM of AcrIF9, which is a concentration of AcrIF9 allowing the occupation of all AcrIF9 binding sites on the Csy complexes. This indicates that two AcrIF9 per Csy complex are required to bind non-specific dsDNA.

Is this compatible with the 3D reconstruction showing two non-contiguous dsDNAs bound to the Csy-AcrIF9 complex?

7) Page 5: "The model reveals that the dsDNA segments are interacting with a positive patch on AcrIF9. [...] Potential interactions are formed between dsDNA and the Csy complex (Fig. 2 c-d), however the resolution does not allow for reliable model building or structure-guided mutagenesis."

- The authors have showed that AcrIF9 alone does not bind to dsDNA. They have also identified potential interactions between the Csy complex and dsDNA in the 3D reconstruction of the ternary Csy-AcrIF9-dsDNA complex. These results indicate that AcrIF9 and Csy surfaces both contribute to the dsDNA binding interfaces. Even though the resolution of the Csy-AcrIF9-dsDNA 3D reconstruction does not allow the modeling of amino acid side chains, the rigid and flexible fitting of the Csy-AcrIF9 model within the Csy-AcrIF9-dsDNA 3D reconstruction may enable the mapping of Csy positively charged residues and the identification of a positive patch at the Csy-AcrIF9 interfaces accommodating the dsDNAs. The authors have also identified some AcrIF9 basic residues that mediate DNA binding using the Csy-AcrIF9 model.

8) Page 6: "Collectively, the structures and biochemistry presented here reveal a two-pronged mechanism for AcrIF9 mediated immune suppression."

- Based on the presented data, "suggest" would be more appropriate than "reveal".

9) Page 7: Data availability

- The authors present a pseudo-atomic model of the AcrIF9-Csy-dsDNA complex in the Fig 2, but this model has not been deposited in the PDB. A Ca backbone model of this ternary complex should be made available.

Minor comments

1) References list

- References 12 to 16 are not called in the text.

2) Page 3: "[...] anti-parallel beta sheet cradling [...]", and page 4: "[...] a β -sheet on [...]"

- Check the spelling (greek versus latine letters).

3) Page 4: "[...] accounting for ~30% of the total solvent accessible surface area [...]"

- Is this the surface area of AcrIF9 or of the thumb of Cas7f?

4) Page 4: "The two AcrIF9 binding sites are nearly identical."

- "Similar" instead of "nearly identical".

5) Page 4: "Residues interacting with each of the AcrIF9 molecules superimpose with an r.m.s.d. of 1.4 Å (see methods)."

- Is this the rmsd for equivalent Ca atoms or all backbone atoms?

6) Fig 1

- Add in the legend what the yellow dotted lines represent.

Reviewers' comments:

Reviewer #1 (Remarks to the Author):

This short paper reports on the mechanism of the anti-CRISPR AcrIF9 (F9), which inhibits the type I-F (Csy) effector. This Acr was one of the first identified but has not been characterised structurally or mechanistically. This is achieved here using a combination of cryo-EM and some basic biochemical analyses. The Cryo structure reveals that AcrIF9 binds to the type I-F complex as two monomers by interacting with the Cas7 subunit in a manner that will block target RNA search and binding – a mechanism remarkably similar to AcrIF1.

As judged by EMSA, AcrIF9 does not bind dsDNA on its own up to a concentration of 3 μ M, but does inhibit Csy binding to target DNA when present at 20 nM concentration (ten-fold less than the Csy concentration) – see point 3 below. However, at higher stoichiometries, F9 no longer prevents binding to DNA, which is also difficult to explain.

A cryo structure was also determined for the complex incubated with dsDNA, revealing binding of the DNA to the surface of F9, allowing mapping of the interface as a basic area of F9 with conserved basic residues (see point 5). The authors speculate that F9 functions partly by sequestering Csy on random DNA as well as blocking target DNA binding.

Overall there is some decent cryo-EM data here paired with some preliminary, contradictory and variable quality EMSA data. This leaves any conclusion beyond “F9 probably functions like F1” to be on rather shaky ground.

Response: Thanks for the critical read. We agree that the biochemistry is apparently misleading. However, we see no alternative explanation or contradictions for AcrIF9-mediated inhibition of crRNA-guide hybridization to a DNA target. We attempt to clarify the apparent contradiction in the biochemistry with the two points below and in the revised manuscript.

First, in our EMSA at low concentrations of AcrIF9, we see inhibition of Csy binding to target DNA. AcrIF9 has a similar binding site to what we observed previously for F1, and both Acrs physically block access to specific regions of the crRNA-guide. While the folds for AcrIF1 and AcrIF9 are distinct, it seems clear that they both sterically block crRNA-DNA interactions. Moreover, these results are consistent with those in the recently published PNAS paper by Zhang et al.

Second, at high concentrations of AcrIF9, we see binding of both target and non-target DNA to the Csy-AcrIF9 complex. Unlike F1, F9 triggers a non-sequence specific association with DNA, which restores DNA binding, but formation of the Csy-AcrIF9-dsDNA complex does not rely on base pairing. A structure of the Csy-AcrIF9-dsDNA complex reveals that the dsDNA is bound to a positive patch on AcrIF9. Bound in this fashion the target-DNA will not trigger recruitment of Cas2/3 and will not be degraded, which is consistent with the “cleavage assays” presented in Zhang et al., but the nature of their assay (i.e. detection of cleavage not binding) prevented them determining the mechanistic basis of this result.

Anti-CRISPR-induced non-sequence specific DNA associations have previously been observed for AcrIIA11 and Cas9 (Forsberg et al.), but structural information explaining the interaction is as yet unavailable. We are not suggesting that the mechanism for non-specific DNA association is conserved among different systems, but rather that this phenomenon may prove to be more general than has been previously appreciated.

Specific points:

Point 1. AcrIF9 is described as a 5-stranded anti-parallel beta sheet cradling an alpha helix, but no structural comparison is offered. Are there any similar folds in PDB? Can the structure be compared more fully to AcrIF1, given their striking convergence in binding sites? Can the two binding sites be compared, beyond saying they are “similar”.

Response: A DALI search revealed many structural homologs to AcrIF9 with significant Z-scores. Most of these structural homologs possessed domains with alpha-beta plait or alpha-beta roll topologies. These topologies are common superfolds for small domains (<100 residues) found in proteins with a broad array of functions and evolutionary origins ([doi:10.1016/j.str.2009.06.015](https://doi.org/10.1016/j.str.2009.06.015)). Therefore, structural homology to AcrIF9 is unlikely to indicate shared functionality and we elected not to include this analysis in the manuscript. AcrIF1 and AcrIF9 have overlapping but not identical binding sites. To illustrate these differences, we have added text to the manuscript and edited Supplementary Fig. 2. Figure 1 and Supplementary Fig. 2 provide a full comparison of the structures and binding sites for AcrIF1 and AcrIF9.

On page 3:

“AcrIF9 is a 7.9kDa protein made up of a five stranded, anti-parallel beta sheet cradling an alpha helix (Fig. 1b), a fold notably distinct from AcrIF1 (Supplementary Fig. 2).”

Supplementary material page 10:

“Supplementary Fig. 2: Comparison of AcrIF9 and AcrIF1. a-b. Schematic of the Csy-AcrIF9 (a) and Csy-AcrIF1 (b) complexes, with subunits colored as in Fig. 1. While AcrIF9 interacts with a single Cas7f, AcrIF1 interacts with residues from two neighboring Cas7f molecules. Inter-subunit interactions are denoted by lines. **c-d.** Model of the Csy-AcrIF9 complex (c) and Csy-AcrIF1/AcrIF2 complex (d), Csy subunits shown as pipes and planks, Acrs shown in surface representation. **e-f.** The binding site of AcrIF9.2 (e) and AcrIF9.1 (f), residues interacting with the Acr highlighted in dark blue. AcrIF9 is composed of a five-stranded anti-parallel beta sheet, cradling an alpha helix. AcrIF1 is composed of a four-stranded anti-parallel beta sheet, flanked on one side by two alpha helices. The folds of the Acrs are notably different. In order to illustrate the difference between Acr binding sites, AcrIF1.2 is shown transparent in e. and AcrIF9.2 is shown transparent in f. Cas7f subunits shown in surface representation, crRNA and Acrs in cartoon representation. **g-h.** Detailed view of the interaction interface in the region of the Cas7.6f thumb helix for AcrIF9 (g) and AcrIF1 (h). While the AcrIF9 and AcrIF1 binding sites overlap, the majority of interactions with Csy residues are different, only interactions with S89 and N94 are common to both Acrs. Model shown in cartoon representation, interacting residues shown as sticks, hydrogen bonds are indicated by yellow dashes, hydrophobic interactions by blue dashes.”

Point 2. The “triplicate” EMSAs shown in figure S3 are strikingly different from one another to a surprising degree. For example, the third gel in S3b is quite fundamentally different from the first gel. In the second and third replicates it looks like there is a significant proportion of shifted material half way up the gel over a wide protein range, suggesting an intermediate species, but the gel in figure 2 doesn’t conform to this. These experiments should be repeated to give more confidence in the data and their interpretation. Ideally, an alternative technique such as fluorescence polarization should be considered, as binding or displacement of Csy will give rise to a large anisotropy change using a method that measures true equilibria.

Response: We thank the reviewer for this important point. The previous version of this paper included an outlier. We have now standardized conditions of the EMSA experiments and have resolved these issues.

The intermediate species could be caused by dissociation of dsDNA alone or in combination with a subset of Csy components. However, we do not observe indications of Csy components dissociating in size exclusion chromatography or cryo-electron microscopy, so the latter explanations seem less likely. While we intend to further investigate the exact nature of this species, the conclusions we draw from these EMSA are not altered by the outcome. The data presented here show that AcrIF9 causes the Csy-AcrIF9 complex to bind dsDNA in a non-specific manner and according to our cryo-EM data, the DNA remains uniformly bound to the complex in solution.

Point 3. Figure 2a suggests that one tenth molar ratio of AcrIF9 almost completely blocks binding of Csy to target DNA. Given that at least 90% of the Csy complex should be free and available to bind DNA (assuming non-cooperative binding of AcrIF9), what is the authors’ view of the mechanism here? Notably, in the replicates, this effect is not observed at 20 nM F9 to nearly the same extent, and in the middle gel of S3a there appears to be 50:50 binding to target DNA at a molar ratio of 1:2, which could make more sense.

Response: We agree with the reviewer and have now replaced the outlier EMSA in Figure 2a.

On page 7:

Point 4. It would help to understand how Csy binds to target and non-target DNA in the absence of F9. Although this has no doubt been published previously it would be helpful to show this data for the DNA species used here. It is important to understand how F9 affects DNA binding for both target and non-target DNA.

Response: As the reviewer suggests, the interaction between Csy and target DNA has previously been described in detail (Rollins et al. 2019, Guo et al. 2017, Rollins et al. 2015). We have added references to these papers. Additionally, we show how Csy interacts with target and non-target DNA oligonucleotides in absence of AcrIF9 by EMSA in the first lane of each gel (Figure 2a and Supplemental Figure 3a and 3b). We show that in the absence of AcrIF9, Csy forms a complex with target DNA, while non-target DNA does not associate with Csy.

On page 6:

“This result suggests that formation of the ternary complex (i.e., Csy-AcrIF9-dsDNA) does not rely on base-pairing interactions, and is fundamentally different than target DNA hybridization to the crRNA guide (Rollins et al. 2015, Guo et al. 2017, Rollins et al. 2019).”

Point 5. A binding interface for F9 on dsDNA is observed that does not involve the Csy complex. The text states that “we attempted to make structure-guided mutations of these residues, but the F9 mutants failed to bind the Csy complex”. Leaving aside the observation that there is no data in the paper to support this (were all 5 made? Did they purify as for the wild-type protein?), this is another puzzling observation as the binding interface between F9 and Csy has not been modified.

Response: We designed and purified three mutants (i.e., AcrIF9^{K31A,R32A,K36A}, AcrIF9^{K31E,R32E} and AcrIF9^{K31Q,R32Q}). These mutants exhibited altered size exclusion profiles and defects in binding to the Csy complex in EMSA experiments. The following text has been added to page 6 for clarification:

*“We attempted to make structure-guided mutations of these residues (AcrIF9^{K31A,R32A,K36A}, AcrIF9^{K31E,R32E} and AcrIF9^{K31Q,R32Q}). While the mutants express and purify similar to wild-type AcrIF9, the size exclusion profiles are distinct. In addition, the mutants are defective for blocking crRNA-guided interactions with DNA targets (**Supplementary Fig 6**). Collectively, these observations suggest the mutations result in a folding defect that reduces the affinity for Csy.”*

Reviewer #2 (Remarks to the Author):

Hirschi et al. present the cryo-EM structures of Csy-AcrIF9 complex and Csy-AcrIF9-dsDNA complex at ~3.9 Å and 6.9 Å respectively. In combination with biochemical data, the authors show AcrIF9 not only sterically blocks the binding of target DNA to crRNA, but also promotes nonspecific recruitment of dsDNA, potentially sequestering the complex from target DNA. Even without the high-resolution structure of Csy-AcrIF9-dsDNA complex, the biochemical data and the current density map support their conclusions. This paper significantly broadens the current view of the mechanisms for anti-CRISPR proteins mediated immune suppression, and will be of interest in the community given that CRISPR-Cas immunity is of broad interest.

During the revision session, the structure of Csy-AcrF9 complex has been described in a PNAS paper (PNAS first published March 13, 2020 <https://doi.org/10.1073/pnas.1922638117>). With 2.57 Å high-resolution Csy-AcrF9 complex structure, the authors of the PNAS paper proposed binding of AcrF9 to the Csy complex prevents target DNA binding to crRNA, similar to how AcrF1 works, also in line with part of Hirschi et al. 's conclusions. The author may briefly discuss the results in the PNAS paper. Even the high-resolution structure of Csy-AcrF9 complex recently has been reported, as far as I see, the non-sequence specific dsDNA bound to Csy-AcrF9 complex reported in Hirschi et al. paper significantly advances the understanding of how anti-CRISPR proteins work, I still support this paper to be published.

Response: Thanks for the feedback. We are aware of the PNAS paper and we alerted the editor as soon as it was published. We have been informed that according to journal policy our manuscript can in principle still be published as it was already under review at the time. Per the reviewer's suggestion we have now included a reference to this work in our revised manuscript. We appreciate the reviewer's recognition of the added importance of our dsDNA bound structure and we agree that this offers important new insight.

On page 7:

“While this manuscript was under review, Zhang et al. published a Csy-AcrIF9 structure and similarly concluded that AcrIF9 inhibits target-DNA hybridization to the crRNA guide (Zhang et al. 2020). Here, we provide additional insights into AcrIF9 function and show how the Csy-AcrIF9 complex promotes binding of dsDNA in a non-sequence-specific manner.”

Minor comments:

Point 1) Fig. 1b, the color of Cas7.6f thumb label is not in blue, making the label not clear that what it represents.

Response: We agree with the reviewer and have now changed the Cas7.6f label color to blue.

On page 5:

Point 2) The authors should provide detailed information in the method session about how Csy-AcrIF9 and Csy-AcrIF9-dsDNA complexes are assembled.

Response: We have now included a detailed description of the Csy-AcrIF9 and Csy-AcrIF9-dsDNA complex assembly in the methods section on page 2 of the supplemental material:

“Purification of Csy-AcrIF9 complex

Csy genes (Addgene ID: 89232), a synthetic CRISPR (Addgene ID: 89244), and AcrIF9 (GenBank: EEG86164.1 cloned into pCDF-1b) were co-expressed on separate vectors, as described above. The Csy-AcrIF9 complex was purified using methods similar to those described for the Csy complex, except size exclusion was performed using a Superdex 200 26/60 column (GE Healthcare), equilibrated in 20 mM HEPES pH 7.5, 100 mM NaCl, 5% glycerol and 1 mM TCEP.

Purification of Csy-AcrIF9-dsDNA complex

Purified Csy was first incubated with a 4-fold excess of AcrIF9 in buffer (20 mM HEPES pH 7.5, 100 mM NaCl, 5% glycerol and 1 mM TCEP) at 37°C for 15 minutes. Purified non-target dsDNA (described below) was then added at a 2.5-fold excess over Csy and incubated for an additional 15 minutes at 37°C. Csy-AcrIF9-dsDNA complex was then purified using a Superdex 200 10/300 size exclusion column (GE Healthcare) and the sample was concentrated using a 100 KDa MWCO concentrator (Pierce).”

Point 3) By comparing Supplementary Fig.1a and Supplementary Fig.4a, it's confusing about the peaks they choose. The authors might use different columns to purify the complex, are the two major peaks in each file related to each other in these two gel filtration files?

Response: Indeed, columns with the same matrix (i.e., Superdex 200), but different bed volumes were used to purify the complexes in Supplementary Figures 1a and 4a. A Superdex 200 26/600 was used to purify the Csy-AcrIF9 complex, and a smaller Superdex 200 10/300 was used to purify the Csy-AcrIF9-dsDNA complex. The methods have been updated to clarify this difference:

“The Csy–AcrIF9 complex was purified using methods similar to those described in the section Purification of the Csy complex, except size exclusion was performed in 20 mM HEPES pH 7.5, 100 mM NaCl, 5% glycerol and 1 mM TCEP (Superdex 200 26/60, GE Healthcare).”

“Csy–AcrIF9–dsDNA complex was then purified by size exclusion chromatography on a Superdex 200 10/300 column (GE Healthcare) and the sample, was concentrated using a 100 KDa MWCO concentrator (Pierce).”

Point 4) In supplementary Fig, 4a, the label “10, 11, 13, 15” is not clear about what it presents.

Response: We thank the reviewer for pointing this out, we have now corrected Supplemental Figure 4 by labeling the lanes with the elution volume.

On page 12 of the supplemental material:

Reviewer #3 (Remarks to the Author):

The authors present structural and biochemical analyses of the anti-CRISPR protein AcrIF9 that targets the type I-F CRISPR-Cas system. They have determined the 3D structure of AcrIF9 bound to the RNA-guided surveillance complex (Csy) by cryo-EM at an overall resolution of 3.9 Å. Two copies of AcrIF9 bind to the Csy complex at positions reminiscent of the AcrIF1 binding sites. AcrIF9, like AcrIF1, blocks target DNA binding in Electrophoretic Mobility Shift Assays with sub-stoichiometric ratios of AcrIF9 to Csy complex. However, at higher stoichiometric ratios, AcrIF9 induces target and non-target DNA binding to Csy. The 3D reconstruction of the Csy-AcrIF9-dsDNA ternary complex, determined by cryo-EM at an overall resolution of 6.9 Å, shows that AcrIF9 tethers dsDNA to the Csy complex.

As acknowledged by the authors, a similar phenomenon of anti-CRISPR-induced non-specific DNA binding to CRISPR-Cas surveillance complex has previously been reported for AcrIIA11 that targets the type II CRISPR-Cas9 system (<https://doi.org/10.7554/eLife.46540>). Although the manuscript under review presents the first example of a type I anti-CRISPR protein promoting non-specific DNA binding to the Csy complex in vitro, major revisions are required to strengthen the authors findings and provide an advance in understanding anti-CRISPR proteins. In particular, the role for dsDNA binding in AcrIF9's mode of inhibition remains elusive. What would be the benefit of a concentration-dependent two-pronged CRISPR-Cas inhibition mechanism while the inhibition of target DNA binding and cleavage is effective at low concentration of anti-CRISPR protein (sub-stoichiometric to stoichiometric ratios of AcrIF9 to the Csy complex)? Noteworthy, the paper by Zhang K et al. entitled "Inhibition mechanisms of AcrF9, AcrF8, and AcrF6 against type I-F CRISPR-Cas complex revealed by cryo-EM", which has been published in the course of reviewing this manuscript (www.pnas.org/cgi/doi/10.1073/pnas.1922638117), presents a highly detailed cryo-EM structure of AcrIF9 bound to the Csy complex at an overall resolution of 2.6 Å and shows that AcrIF9 functions via the competitive binding to DNA binding sites on Cas7f subunits. Overall, further data interpretation and additional experiments could greatly improve the manuscript. Besides, all methodological details should be provided for a better understanding of the results and for the sake of reproducibility.

Response: Thank you for your feedback. We are aware of the PNAS paper and we alerted the editor as soon as it was published. We have been informed by the editor, that papers published during review will not detract from the impact of a paper under consideration. We have included a reference to the recently published paper in our revised manuscript.

As the referee points out, Acr-mediated non-sequence specific binding has recently been reported for AcrIIA11 and Cas9 (Forsberg et al.), but prior to the work in this manuscript there have been no structural or mechanistic insights into how this is achieved in any of the CRISPR systems. We speculate about the biological significance of this behavior in our paper, but we agree with the referee that more work needs to be done on AcrIIA11, F9 and other Acrs to determine the biological benefits of this behavior.

Major comments

Point 1) Page 3: "To achieve this goal, we co-expressed and purified the crRNA-guided surveillance complex (Csy) with AcrIF9 and determined the structure [...]"

- Here one understands that AcrIF9 together with the crRNA-guided surveillance complex were co-expressed. However, there is no paragraph related to this procedure in the Methods.

Alternatively, if AcrIF9 and the Csy complex have been produced independently, as described in the Methods, and then mixed to make the complex, this procedure should be described in the Methods.

This point should be clarified.

Response: We agree with the reviewer and have now included a description of the Csy-AcrIF9 complex co-expression and purification in the methods section on page 2 of the supplemental material:

“Purification of Csy-AcrIF9 complex

Csy genes (Addgene ID: 89232), a synthetic CRISPR (Addgene ID: 89244), and AcrIF9 (GenBank: EEG86164.1 cloned into pCDF-1b) were co-expressed on separate vectors, as described above. Csy-AcrIF9 complex was purified using methods described above for Csy alone (Superdex 200 26/60 GE Healthcare) and stored in complex buffer (20 mM HEPES pH 7.5, 100 mM NaCl, 5% glycerol and 1 mM TCEP).”

Point 2) Fig 1, panels d-e; page 4: “The two AcrIF9 binding sites are nearly identical. Residues interacting with each of the AcrIF9 molecules superimpose with an r.m.s.d. of 1.4 Å (see methods). The contacts include: a hydrogen bonding network involving AcrIF9 residues 11 to 15, a β-sheet on the Cas7f thumb, and a nucleobase from the crRNA (Fig. 1c); interactions between AcrIF9 arginine 17 and residues in the Cas7f thumb (Fig. 1d); and the formation of an “aromatic clamp” around the Cas7f thumb loop by AcrIF9 residues 39 to 41 (Fig. 1d). AcrIF9.1, but not AcrIF9.2, also interacts with Cas8f (Fig 1e).”

-This description of the intermolecular contacts between AcrIF9 and the Csy complex should be revised for a better reading and understanding.

Response: We thank the reviewer for the suggestion and have now rewritten this section for improved readability. On page 4:

“AcrIF9 is a 7.9kDa protein made up of a five stranded, anti-parallel beta sheet cradling an alpha helix (Fig. 1b), a fold distinct from AcrIF1 (Supplementary Fig. 2). Both AcrIF9 molecules form extensive interactions with the thumbs of Cas7f (Fig. 1c-d), accounting for ~30% of the total solvent accessible surface area of AcrIF9. The two AcrIF9 binding sites are similar, and the specific residues interacting with each of the AcrIF9 molecules superimpose with an r.m.s.d. of 1.4 Å (see methods). The contacts include a hydrogen bonding network between AcrIF9 residues Q11 to S15, and residues of Cas7f; N94 and Q96 in the Cas7f thumb beta sheet, S262 in the Cas7f extended webbing and a nucleobase from the crRNA (C13 and U25 for AcrIF9.1 and AcrIF9.2 respectively, Fig. 1c). A second cluster of hydrogen bonding interactions is found between AcrIF9 R17 and residues S89 and S92 in the Cas7f thumb (Fig. 1d). Furthermore, the aromatic sidechains of AcrIF9 residues F39, F40 and H41 form an “aromatic clamp” around the Cas7f thumb loop residues L76 to T78 and additional hydrogen bonding occurs between AcrIF9 residues Q38 to H41 and Cas7f residues R75 to T78 (Fig. 1d). AcrIF9.1, but not AcrIF9.2, also interacts with Cas8f through hydrogen bonding of the Cas8f R224 carbonyl with AcrIF9.1 L27 and T28 backbone nitrogens (Fig 1e). While additional interactions between AcrIF9.1 and Cas8f are expected, the local resolution limits confident assignment of additional interactions at this interface.”

- The authors mention an “aromatic clamp” formed by AcrIF9 residues, which suggests hydrophobic or stacking interactions between AcrIF9 and Cas7f residues. However, panel d shows that these aromatic residues hydrogen bond with Cas7f.

Response: An “aromatic clamp” is formed by the sidechains of AcrIF9 residues F39, F40 and H41; the aromatic rings of these residues make hydrophobic interactions with the backbone of

Cas7.6 thumb loop residues R75, L76 and K77. Additionally, backbone atoms of these residues are involved in hydrogen bonding. We have now included a detailed description of each individual interaction and a graphical indication of hydrophobic interactions in Figure 1. On page 5:

“Fig 1: Cryo-electron microscopy structure of the Csy-AcrIF9 complex. a. Model of the Csy-AcrIF9 complex, Csy subunits shown as pipes and planks, AcrIF9 shown in surface representation. **b.** The binding site of AcrIF9.2, residues interacting with AcrIF9 highlighted in blue (Cas7.6f) and green (crRNA). Cas7f subunits shown in surface representation, crRNA and AcrIF9 in cartoon representation. **c-e.** Detailed view of the AcrIF9 binding site, model shown in cartoon representation, interacting residues shown as sticks. Interactions with uridine 25 (U25) are expected to accommodate any base at the equivalent position. Hydrogen bonds are indicated by yellow dashes, hydrophobic interactions are indicated by blue dashes.”

- The authors mention “interactions between AcrIF9 arginine 17 and residues in the Cas7f thumb”. The term “interactions” is vague. According to panel d, these interactions are hydrogen bonds, as are all intermolecular contacts described in this section.

Response: We have now edited the text in order to specify that the interactions with arginine 17 are indeed hydrogen bonding interactions, please see above for the edited text describing the interactions.

- The authors indicate that the two AcrIF9 binding sites are nearly identical. Is the only

difference the interaction with Cas8f? This interaction between Cas8f and AcrIF9.1, shown as one hydrogen bond in panel e, should also be described in the text.

Response: Indeed, the only difference in the two binding sites is the interaction with Cas8f, which exists only for AcrIF9.1. Unfortunately, the reconstruction is not well-resolved in the region of this interaction which limits modeling of Cas8f sidechains that are potentially involved in further contacts. The only hydrogen bond that we can confidently report occurs between the backbones of AcrIF9 L27 and Cas8f R224. We have edited the text to clarify this point. Please see above for the edited text describing the interactions.

- The numbering of all interacting residues and nucleobases should be reported in the text and the figure panels.

Response: We have revised the text and figure panels according to the referee's suggestion.

Point 3) Page 5: "Based on the similarity of the binding sites for AcrIF1 and AcrIF9, we predicted that AcrIF9 would sterically block target binding."

-The authors have previously shown that AcrIF1 interacts with Cas7f lysine residues (K85, K254, K257) that are involved in target DNA binding (<http://dx.doi.org/10.1016/j.cell.2017.03.012>). Based on Fig 1 panel d, the lysine 85, which is located in the thumb of Cas7f, may also interact with AcrIF9. Testing the binding of AcrIF9 to wild type Csy and to the K85A mutant used in the paper mentioned above, could experimentally prove that AcrIF9 is a steric inhibitor of target DNA binding.

Response: AcrIF9 does not interact with Cas7f residues K85, K254 or K257. While the binding sites for AcrIF1 and AcrIF9 are similar, they are not identical. In order to clarify these differences, we have added panels to Supplementary Figure 2.

Supplementary material page 10:

“Supplementary Fig. 2: Comparison of AcrIF9 and AcrIF1. a-b. Schematic of the Csy-AcrIF9 (a) and Csy-AcrIF1 (b) complexes, with subunits colored as in Fig. 1. While AcrIF9 interacts with a single Cas7f, AcrIF1 interacts with residues from two neighboring Cas7f molecules. Inter-subunit interactions are denoted by lines. **c-d.** Model of the Csy-AcrIF9 complex (c) and Csy-AcrIF1/AcrIF2 complex (d), Csy subunits shown as pipes and planks, Acrs shown in surface representation. **e-f.** The binding site of AcrIF9.2 (e) and AcrIF9.1 (f), residues interacting with the Acr highlighted in dark blue. AcrIF9 is composed of a five-stranded anti-parallel beta sheet, cradling an alpha helix. AcrIF1 is composed of a four-stranded anti-parallel beta sheet, flanked on one side by two alpha helices. The folds of the Acrs are notably different. In order to illustrate the difference between Acr binding sites, AcrIF1.2 is shown transparent in e. and AcrIF9.2 is shown transparent in f. Cas7f subunits shown in surface representation, crRNA and Acrs in cartoon representation. **g-h.** Detailed view of the interaction interface in the region of the Cas7.6f thumb helix for AcrIF9 (g) and AcrIF1 (h). While the AcrIF9 and AcrIF1 binding sites overlap, the majority of interactions with Csy residues are different, only interactions with S89 and N94 are common to both Acrs. Model shown in cartoon representation, interacting residues shown as sticks, hydrogen bonds are indicated by yellow dashes, hydrophobic interactions by blue dashes.”

Point 4) Page 6: Fig 2, panel A

- In the EMSA with target dsDNA, the strong unbound DNA band in the 20 nM lane is puzzling. Indeed, it indicates that most of the Csy molecules are inhibited, which cannot be the case with a steric inhibitor at sub-stoichiometric concentration. The second EMSA presented in the Supplementary Fig 3 panel a would be more appropriate for the main figure.

Response: We thank the reviewer for pointing this out and acknowledge the importance of this issue. Unfortunately, we previously included an outlier EMSA in Figure 2. The EMSA has been optimized and these conditions have been used to perform the experiments in triplicate.

On page 7:

- In the EMSA with target and non-target dsDNAs, how do you interpret the DNA migration profiles of the 200 nM lanes? At 200 nM of AcrIF9, and with two AcrIF9 binding sites per Csy molecule, not all Csy molecules are fully occupied by the inhibitor.

Response: We expect that the smear in the ≥ 100 nM AcrIF9 lanes is caused by P32-labeled dsDNA dissociating from the complex. Alternatively, this band could represent an intermediate complex that comprises dsDNA and a subset of Csy components. However, we do not observe indications of such complexes in size exclusion chromatography or cryo-electron microscopy, and thus this seems a less likely explanation. We intend to further investigate the exact nature of the complex causing the smear. However, the conclusions we currently draw from the EMSA are not affected by the outcome of these investigations. These data show that AcrIF9 causes the Csy-AcrIF9 complex to bind dsDNA in a non-specific manner.

- Overall, how do you interpret these EMSA in terms of the AcrIF9 molecular mechanism? If a Csy-mediated DNA cleavage assay in the presence of a stoichiometric ratio of AcrIF9 shows an inhibition of the Csy activity (which is not the case for AcrIIA11), what would be the benefit of binding dsDNA at high concentrations of AcrIF9?

Response: In the paper, we suggest that AcrIF9-induced binding to non-sequence specific dsDNA may provide an additional layer of immune suppression by sequestering Csy in non-productive associations with dsDNA. However, we cannot rule out that non-specific DNA binding capacity serves an alternative purpose (e.g. Csy has been implicated in new sequence adaptation). Determining the biological impact of Acr-mediated non-sequence specific sequestration is part of our future research program.

Point 5) Page 5: “we next determined the structure of the Csy-AcrIF9 complex bound to dsDNA at a nominal resolution of ~6.9 Å (Fig. 2b and Supplementary Fig. 4).”

- The procedure used to make the Csy-AcrIF9-dsDNA ternary complex is not described in the Methods.

Response: We thank the reviewer for pointing this out and have added a section to describe the procedure in the supplemental material on page 2:

“Purification of Csy-AcrIF9–dsDNA complex

Purified Csy was first incubated with a 4-fold excess of AcrIF9 in complex buffer at 37°C for 15 minutes. Purified non-target dsDNA (prepared as below) was then added at a 2.5-fold excess over Csy and incubated for an additional 15 minutes. at 37°C. Csy–AcrIF9–dsDNA complex was then purified using a Superdex 200 10/300 size-exclusion column (GE Healthcare). The high molecular weight complex eluting before the peak corresponding to Csy alone, was concentrated in a 100 KDa MWCO concentrator (Pierce).”

Point 6) Page 5: “The reconstruction reveals two stretches of non-contiguous helical density that are not accounted for by Csy or AcrIF9, but are consistent with the size and shape of B-form DNA (Fig. 2c).”

-The EMSA with target and non-target dsDNAs show that non-specific DNA binding occurs at 400 nM of AcrIF9, which is a concentration of AcrIF9 allowing the occupation of all AcrIF9 binding sites on the Csy complexes. This indicates that two AcrIF9 per Csy complex are required to bind non-specific dsDNA.

Is this compatible with the 3D reconstruction showing two non-contiguous dsDNAs bound to the Csy-AcrIF9 complex?

Response: Results from the EMSA are compatible with the 3D reconstruction showing two non-contiguous dsDNAs bound to the Csy-AcrIF9 complex. We found non-specific DNA binding starting to occur at 100 nM AcrIF9, at which point we expect Csy-AcrIF9 complexes to have formed. Non-specific DNA binding reaches a plateau at ~700-1000 nM AcrIF9, when we expect most Csy complexes to be occupied by two AcrIF9 molecules.

Point 7) Page 5: “The model reveals that the dsDNA segments are interacting with a positive patch on AcrIF9. [...] Potential interactions are formed between dsDNA and the Csy complex (Fig. 2 c-d), however the resolution does not allow for reliable model building or structure-guided mutagenesis.”

- The authors have showed that AcrIF9 alone does not bind to dsDNA. They have also identified potential interactions between the Csy complex and dsDNA in the 3D reconstruction of the ternary Csy-AcrIF9-dsDNA complex. These results indicate that AcrIF9 and Csy surfaces both contribute to the dsDNA binding interfaces. Even though the resolution of the Csy-AcrIF9-dsDNA 3D reconstruction does not allow the modeling of amino acid side chains, the rigid and flexible fitting of the Csy-AcrIF9 model within the Csy-AcrIF9-dsDNA 3D reconstruction may enable the mapping of Csy positively charged residues and the identification of a positive patch at the Csy-AcrIF9 interfaces accommodating the dsDNAs. The authors have also identified some AcrIF9 basic residues that mediate DNA binding using the Csy-AcrIF9 model.

Response: We have improved the resolution of the Csy-AcrIF9-dsDNA structure to ~4.2 Å, which will help guide future mutational studies. However, we are currently unable to perform

further mutagenesis and binding experiments due to mandated shutouts intended to reduce the spread of SARS-CoV2. We have been given no guidance on how long the shutout will persist.

Point 8) Page 6: “Collectively, the structures and biochemistry presented here reveal a two-pronged mechanism for AcrIF9 mediated immune suppression.”

- Based on the presented data, “suggest” would be more appropriate than “reveal”.

Response: We agree with the reviewer and edited the text accordingly, on page 7:

“Collectively, the structures and biochemistry presented here suggest a two-pronged mechanism for AcrIF9 mediated immune suppression.”

Point 9) Page 7: Data availability

- The authors present a pseudo-atomic model of the AcrIF9-Csy-dsDNA complex in the Fig 2, but this model has not been deposited in the PDB. A C α backbone model of this ternary complex should be made available.

Response: We have been able to improve the resolution of the dsDNA bound structure. We have deposited the model in the PDB under the following coordinates: 6WHI.

Minor comments

Point 1) References list

- References 12 to 16 are not called in the text.

Response: We have now removed the references.

Point 2) Page 3: “[...] anti-parallel beta sheet cradling [...]”, and page 4: “[...] a β -sheet on [...]”

- Check the spelling (greek versus latine letters).

Response: We have now edited the text for consistency in the spelling of beta-sheet.

Point 3) Page 4: “[...] accounting for ~30% of the total solvent accessible surface area [...]”

- Is this the surface area of AcrIF9 or of the thumb of Cas7f?

Response: We now added a clarification for the nature of the surface area, on page 3-4:

“Both molecules of AcrIF9 form extensive interactions with the thumbs of Cas7f (Fig. 1c-d), accounting for ~30% of the total solvent accessible surface area of AcrIF9.”

Point 4) Page 4: “The two AcrIF9 binding sites are nearly identical.”

- “Similar” instead of “nearly identical”.

Response: We have now edited the text as follows, on page 4:

“The two AcrIF9 binding sites are similar, residues interacting with each of the AcrIF9 molecules superimpose with an r.m.s.d. of 1.4 Å (see methods).”

Point 5) Page 4: “Residues interacting with each of the AcrIF9 molecules superimpose with an r.m.s.d. of 1.4 Å (see methods).”

- Is this the rmsd for equivalent C α atoms or all backbone atoms?

Response: The r.m.s.d was calculated for all atoms of the listed residues. We have now clarified the point as follows in the supplementary data page 7:

“In order to compare the AcrIF9 binding sites, the AcrIF9 molecules were superimposed using Pymol followed by calculation of the all atom r.m.s.d. between the C α residues within 4 Å from AcrIF9.1 or AcrIF9.2.”

Point 6) Fig 1

- Add in the legend what the yellow dotted lines represent.

Response: We agree with the reviewer and added the explanation in the figure legend as follows:

“Hydrogen bonds are indicated by yellow dashes, hydrophobic interactions are indicated by blue dashes.”

REVIEWERS' COMMENTS:

Reviewer #2 (Remarks to the Author):

In this manuscript, Hirschi et al. describe the cryo-EM structures of Csy-AcrIF9 complex and Csy-AcrIF9-dsDNA complex. Under structural and biochemical data, the authors show AcrIF9 not only sterically blocks the binding of target DNA to crRNA, but also promotes nonspecific recruitment of dsDNA, potentially sequestering the complex from target DNA. The authors have addressed all my concerns by adding new text during revision.

Reviewer #3 (Remarks to the Author):

The comments have been addressed by the authors.
However, should the EMSA protocol in the Methods not be updated with the optimized conditions presented in the revised manuscript?

Response to Review

Reviewer's comments:

Reviewer #2 (Remarks to the Author):

In this manuscript, Hirschi et al. describe the cryo-EM structures of Csy-AcrIF9 complex and Csy-AcrIF9-dsDNA complex. Under structural and biochemical data, the authors show AcrIF9 not only sterically blocks the binding of target DNA to crRNA, but also promotes nonspecific recruitment of dsDNA, potentially sequestering the complex from target DNA. The authors have addressed all my concerns by adding new text during revision.

Reviewer #3 (Remarks to the Author):

The comments have been addressed by the authors. However, should the EMSA protocol in the Methods not be updated with the optimized conditions presented in the revised manuscript?

Response:

The methods were updated, but this change had not been highlighted with track changes in the previous submission. The corrected methods, read as follows:

“Binding assays were performed by incubating increasing concentrations of AcrIF9 with 200 nM Csy complex in reaction buffer (20 mM HEPES pH 7.5, 100 mM potassium acetate, 5% glycerol, 1 mM TCEP) for 15 minutes at 37°C.”